# Development of a Screening Platform for Optimizing Chemical Nanosensor Materials

**DOI:** 10.3390/s24175565

**Published:** 2024-08-28

**Authors:** Larissa Egger, Lisbeth Reiner, Florentyna Sosada-Ludwikowska, Anton Köck, Hendrik Schlicke, Sören Becker, Öznur Tokmak, Jan Steffen Niehaus, Alexander Blümel, Karl Popovic, Martin Tscherner

**Affiliations:** 1Microelectronics, Materials Center Leoben Forschung GmbH, 8700 Leoben, Austria; lisbeth.reiner@gmail.com (L.R.); florentyna.sosada-ludwikowska@mcl.at (F.S.-L.); anton.koeck@mcl.at (A.K.); 2Institute of Physical Chemistry and Polymer Physics, Leibniz Institute of Polymer Research Dresden, 01069 Dresden, Germany; 3Fraunhofer Institute for Applied Polymer Research IAP, Center for Applied Nanotechnology CAN, 20146 Hamburg, Germanyoeznur.tokmak@iap.fraunhofer.de (Ö.T.); jan.steffen.niehaus@iap.fraunhofer.de (J.S.N.); 4Joanneum Research, Institute for Surface Technologies and Photonics, 8160 Weiz, Austria; alexander.bluemel@joanneum.at (A.B.); karl.popovic@joanneum.at (K.P.); martin.tscherner@joanneum.at (M.T.)

**Keywords:** nanosensors, hybrid nanomaterials, metal oxide gas sensors, nanoparticles

## Abstract

Chemical sensors, relying on changes in the electrical conductance of a gas-sensitive material due to the surrounding gas, typically react with multiple target gases and the resulting response is not specific for a certain analyte species. The purpose of this study was the development of a multi-sensor platform for systematic screening of gas-sensitive nanomaterials. We have developed a specific Si-based platform chip, which integrates a total of 16 sensor structures. Along with a newly developed measurement setup, this multi-sensor platform enables simultaneous performance characterization of up to 16 different sensor materials in parallel in an automated gas measurement setup. In this study, we chose the well-established ultrathin SnO_2_ films as base material. In order to screen the sensor performance towards type and areal density of nanoparticles on the SnO_2_ films, the films are functionalized by ESJET printing Au-, NiPt-, and Pd-nanoparticle solutions with five different concentrations. The functionalized sensors have been tested toward the target gases: carbon monoxide and a specific hydrogen carbon gas mixture of acetylene, ethane, ethne, and propene. The measurements have been performed in three different humidity conditions (25%, 50% and 75% r.h.). We have found that all investigated types of NPs (except Pd) increase the responses of the sensors towards CO and HC_mix_ and reach a maximum for an NP type specific concentration.

## 1. Introduction

Chemical sensing of gaseous molecules has become a vital necessity for a variety of applications ranging from indoor and outdoor air quality and atmospheric pollution monitoring to industrial workplace safety and the detection of toxic and explosive gases. Conductometric gas sensors rely on changes in the electrical conductance of a gas-sensitive material due to the surrounding gas. These devices are typically based on metal oxide (MOx) semiconductors, such as SnO_2_, ZnO, CuO, TiO_2_, or WOx, and are the most promising types of solid-state gas sensors for cost-efficient compact implementations with industrial relevance [1,2,3,4,5,6]. Significant progress has been achieved in optimizing the performance of these sensor devices in terms of sensitivity, power consumption, and miniaturization. Low selectivity, however, remains the biggest disadvantage of common conductometric gas sensors. Typically, metal oxides react with multiple target gases and the resulting changes in conductivity are not specific for a certain analyte species.

The implementation of nanomaterials, such as nanocrystalline thin films, nanowires, or nanoparticles, is a highly powerful strategy to improve sensor selectivity [7,8,9,10]. Much progress has been achieved in optimizing the sensor device properties by tailoring the nanostructure morphology and by multi-component nanomaterial approaches, including surface modification and additive doping [11,12,13,14]. The performance of SnO_2_-based devices, for example, can be enhanced by employing sensing elements based on heterostructures [15,16,17,18] or by the addition of noble metals as catalysts to the MOx [19,20]. Tailoring the response of MOx sensors by surface functionalization with metallic nanoparticles, such as Au, Pd, or Pt, is a highly promising approach to achieve a high degree of selectivity [21,22,23,24].

Metallic NPs can be fabricated by different physical and chemical technologies. Physical methods encompass sputtering techniques, such as magnetron sputtering, evaporation methods, spark ablation, or laser ablation from bulk metal targets. Recently, we have employed a magnetron sputtering-based gas-phase synthesis approach to functionalize SnO_2_ ultrathin films, with preformed Pt-NPs exhibiting typical sizes in the range of 1–5 nm. When coated with Pt-NPs with an average diameter below 2 nm, the CMOS integrated sensor devices exhibit significantly enhanced carbon monoxide (CO) detection with minimized humidity interference [25]. Recently, it has been shown that Pt nanocatalyst decoration can be applied for selectivity control in CMOS integrated SnO_2_ thin film gas sensors. [26]

The response of NP-functionalized sensors is determined by the type, size, and shape of the NPs as well as by the aerial density (i.e., number of NPs per square unit). Physical techniques require sophisticated equipment and are commonly limited in terms of material composition, and control over size and shape, resulting in a broad size distribution of the fabricated NPs. An alternative is the solution-based colloidal synthesis of NPs, where wet-chemical methods are used to grow NPs in solution from molecular precursor molecules. The adjustment of reaction conditions, such as temperatures and precursor concentration, as well as the proper choice of precursor molecules, enables precise tuning of the properties (size, shape, composition) of the obtained NPs. The NPs can be introduced into ink formulations suitable for the solution-based functionalization of MOx sensor surfaces. This provides a high degree of flexibility and tuneability of the sensor properties.

The purpose of this study was the development of a multi-sensor platform for systematic screening of gas-sensitive nanomaterials. We have developed a specific Si-based platform chip, which integrates a total of 16 sensor structures. In this paper, we demonstrate the utility of the platform chip for a systematic screening of the most prominent and promising gas sensing nanomaterials reported in the literature: SnO_2_, and three different types of metallic NPs (Au [27,28,29], Ni_0.3_Pt_0.7_ [30], Pd [31,32]). We employ ultrathin SnO_2_-films (thickness: 50 nm) which are deposited by our own spray pyrolysis technology [33]. This tool enables deposition of SnO_2_, ZnO, and CuO films over a full 200 mm (diameter) wafer, which is of high importance to upscale our sensor technology [34]. We have chosen well-established ultra-thin SnO_2_ films as base material and Au, Ni_0.3_Pt_0.7_, and Pd as NPs for functionalization, because we have performed preliminary experimental studies with these types of NPs, which are synthesized with high reproducibility by the partner Fraunhofer IPA-CAN. We have developed a 2 × 2 cm^2^ Si-based platform chip, which integrates a total of up to 16 different nanomaterials. The sensor structures have been functionalized by ESJET printing of NP-solutions with five different concentrations in order to study the dependence of the sensor response on the NPs’ aerial density. The platform chip is read out by a newly developed automated gas measurement setup, which enables simultaneous measurements of all 16 sensors in parallel [35]. The NP-functionalized sensors have been tested toward the target gases CO and a specific hydrogen carbon gas mixture (=HC_mix_) of acetylene, ethane, ethene, and propene. To demonstrate the usability of our platform chips for sensor screening, we have focused on 50 ppm concentrations for both test gases and an operation temperature of 300 °C. In the literature, it was shown that these parameters are representative and promising sensing parameters for SnO_2_-based gas sensors. [36,37,38,39]. The measurements have been performed in three different humidity conditions (25%, 50% and 75% r.h.) to mimic real life conditions. We have found that all investigated types of NPs (except Pd) increase the responses of the sensors towards CO and HCMix and reach a maximum for an NP type specific concentration.

## 2. Materials and Methods

### 2.1. Platform Screening Chip and Gas Measurement Setup

The specific platform chip, which was designed and fabricated to enable efficient screening of up to 16 various sensor materials on a single sensor chip in a single measurement run, is shown in Figure 1. The platform chips are fabricated on a full 200 mm Si wafer size by photolithography, Ti/Pt-sputtering and lift-off. The wafer was afterwards diced into a 2 × 2 cm^2^ sized Si chips (thickness 600 µm, plus 100 nm SiO_2_ coating). The chip integrates 66 Ti/Pt-contact pads (size 150 × 200 µm^2^, thickness: 50 nm, 150 nm), respectively. Ti/Pt metal lines lead from the contact pads to the 16 sensors located in the “hot” middle part of the chip, which is heated during sensor performance measurements (see Figure 2). For sensor fabrication, a nanocrystalline SnO_2_ film is deposited on the whole platform chip by a low-cost spray pyrolysis technology performed at atmospheric pressure with a thickness of around 50 nm [8]. Subsequently, the SnO_2_ film is structured utilizing photolithography and ion etching; the photoresist (AZ MIR 701) is removed by a proper remover (AZ Remover). A final 50 × 100 µm^2^ sized SnO_2_ sensor structure is shown in the insert of Figure 1. The electrodes are designed to provide conductive measurement of the SnO_2_-film in a 4-point measurement configuration; thus, each sensor requires 4 contact pads. Similar to a Pt100 temperature sensor structure, meandering Ti/Pt metal lines are additionally integrated on the platform chip to calibrate the temperature on the chip. This requires two more contact pads: each platform chip exhibits a total of 2 × 33 + 2 = 66 contact pads.

A specific gas test chamber has been designed and fabricated to characterize the sensor performance. Figure 2a shows the whole measurement setup in the open state. The setup consists of a holder for the platform chip, a lid containing the gas inlet, the required electrical connectors, the contact pins, and a sealing. Figure 2b shows a front view of the lid, which holds two prober heads, each of them containing 33 contact pins. Figure 2c shows in more detail the chip holder: the chip is precisely placed on an Au-coated Cu-heater (size 3 × 1 cm^2^), which heats the central part of the chip (containing the 16 sensor structures) from underneath up to 350 °C.

As shown in [35], we have carefully evaluated the heating performance of our measurement setup up to 550 °C. The heating performance of the Cu-heater and the heat distribution on the platform chip have both been both simulated, as well as experimentally evaluated by thermography. The red rectangle in Figure 1 designates the area of the platform chip heated by the Au-coated Cu-block from underneath. The heat distribution across this area was found to be within ±5 °C as verified by thermography. Immediately out of the “hot zone” in the direction of the contact pads, the temperature drops to ambient air temperature. This excellent homogeneity guarantees the same operation temperature for all sensors, and a variation of ±5 °C is negligible for the sensor response.

When the lid is closed, the pins contact the Ti/Pt contact pads on the platform chip and enable simultaneous characterization of all 16 sensor structures in parallel. The gas is inserted in the middle of the lid and flows over the platform chips down through the openings to the left and right of the chip.

### 2.2. Synthesis of Nanoparticles

Different types of noble metal nanoparticles were fabricated for the functionalization of the SnO_2_ sensing films. Au-NPs were synthesized following a procedure by Peng et al. and Wu et al. with specific modifications [28,29]. Therefore, tetrachloroauric acid was reduced by borane tert-butylamine in the presence of dodecylamine and using hexane as a solvent. In a similar process, Pd-NPs were obtained via a method described by Mazumder et al. [31]. Here, palladium(II) acetylacetonate (Pd(acac)2) was reduced using the same reducing agent in the presence of oleylamine acting as a stabilizer. Finally, Ni_x_Pt_1-x_ NPs (referenced further as NiPt-NPs) were fabricated according to a modification of Wang et al. [30] by reducing a mixture of Platinum(II) acetylacetonate (Pt(acac)2) and Nickel(II) acetylacetonate (Ni(acac)2) via the addition of Fe(CO)_5_ in the presence of oleylamine and oleic acid. The composition of the NiPt nanoparticles was determined via energy-dispersive X-ray analysis (EDX) using a scanning electron microscope (SEM). Therefore, the purified nanoparticles dispersed in hexane were drop cast on a silicon wafer. The resulting nanoparticle film was examined in a Zeiss LEO Gemini 1550 (Potsdam, Germany). The silicon drift detector used was an Oxford Instruments Ultim Max 100 (Abingdon, UK). The composition determined in this way was typically in the range of Ni_0.3_Pt_0.7_. All NPs were purified via precipitation by the addition of non-solvents and redispersed in octane.

Finally, proper inks consisting of NPs dissolved in a mixture of octane/1-octanol (20:80, *v*:*v*) were formulated. The inks additionally contained approximately 5 mg/mL oleylamine to assure particle stability throughout storage and printing. Octane/1–octanol mixtures were chosen as solvents for the NP inks. The characteristics of these inks have proven fully compatible with the high-resolution Electrostatic InkJET technology-based printing process (ESJET), which has been applied for the functionalization of the SnO_2_ sensor films on the Si-platform chips with NPs. To screen the effects of NP functionalization on the sensor responses, inks with all three types of NPs and various concentrations were prepared. The base concentration of the NP inks was ~1.22 µM, and ~1.70 µM ~3.19 µM for the Au, NiPt, and Pd particle systems, respectively. Further, each of these solutions was diluted by factors of 0.5, 0.25, 0.125, and 0.0625, prior to deposition via ESJET printing. Table 1 summarizes the target dilution ratio, the required volumes of the NP base solution, octane, and 1-octanol, respectively, to achieve a total NP-ink volume of 5 mL; also shown are the resulting NP concentrations for the Au, NiPt, and Pd particle systems.

### 2.3. Functionalization of the Sensor Films with Nanoparticle Solutions

The NP solutions have been printed by ESJET technology, which is a proprietary technique, not yet commercialized, from the company Precision Varionic Intl. Ltd., for droplet generation, which is based on a completely different principle than in piezo-based inkjet processes: The ESJET employs natural pulsations usually occurring in electrospray systems, where pulses of an electric field of defined strength and duration are used to achieve the generation of well controlled, very small volumes of both non-conductive as well as conductive liquids. An ESJET system (Figure 3, left) consists of a capillary-shaped emitter with a needle tip electrode positioned inside for applying an electric field or injecting charges into the printing ink. The emitter is positioned a short distance (typically ~1 mm) from a target, which is kept at ground potential. When a voltage pulse is applied, the ink follows the electrostatic forces and is accelerated against the substrate. The voltage pulses (typically 0.9–5 kV), including their temporal dynamics, must be precisely controlled in terms of amplitude, shape and duration to achieve reproducible results. Apart from accurate x-y-z positioning of the emitter above the substrate, the method does not require any further moving parts, such as actuators or micropumps, enabling its easy integration into a printing platform. A customized PixDro LP50 tool (Suss MicroTec, Garching, Germany) was used as positioning unit of the platform chip.

While, in traditional inkjet technology, visco-elastic effects in the ink play a very critical role, the ESJET process can handle a much wider range of rheological ink properties. The rheological properties of the NP ink formulations proved to be ideal for processing by means of ESJET printing. In the first tests, arrays of 100 dots and more with a diameter of about 80 µm could be easily printed. Figure 3, right, shows the nozzle positioned above a single SnO_2_ sensor film on the Si-platform chip; the NP-dot diameter matches the active area of the sensor films, which measures 30 × 50 µm^2^ (see Figure 1). The distribution of the NPs in the area of the printed dots was quite even, and no “coffee ring” effects have been observed.

To provide a different areal density of the NPs on the sensor surface, the NP base solutions were diluted as described in Section 2.2. Fresh NP formulations were prepared from NP base solutions immediately before the printing process for the functionalization of the SnO_2_ sensor films. In order to reveal the influence of the material (Au, Pd, NiPt) and the areal density of the NPs on the sensor response, the NP solutions have been printed in 5 different dilutions (1:1, 1:2, 1:4, 1:8, and 1:16) in a specific scheme on the Si-platform chip, as shown in Figure 4. A total of 10 sensors is functionalized with NPs; two sensors, respectively, per NP-dilution; six SnO_2_ sensor films are not functionalized with NPs and form the references. The printing was performed on slightly heated substrates (80 °C) to accelerate the evaporation of the solvent.

### 2.4. Characterization of Nanoparticles on Sensor Surface

The scanning electron microscopy (SEM) characterization can be performed only after full sensor performance characterization to avoid any surface contamination due to the electron beam, which is detrimental to the sensing performance. Therefore, the only SEM-graphs in this publication were performed after gas measurements. The areal densities of the NPs have been determined by SEM analysis with InLens and ESB detectors. The SnO_2_ films (reference sensors, as well as NP-functionalized sensors) exhibit a rather granular structure, which shows that NPs have no influence on the microstructure of the SnO_2_ base material. Figure 5 shows an SEM graph (ESB detector, 200,000× magnification) of Sensor A, which is functionalized with Au-NPs. The SnO_2_ film exhibits a rather granular structure, and the Au-NPs can be clearly seen as white dots; the NPs seem to be rather uniformly distributed, and no large agglomerations of NPs are visible. The number of NPs has been counted “manually”: Sensor A functionalized with the pure Au-NP ink (dilution ratio 1:1) shows ca. 350 NPs/µm^2^, while Sensor H functionalized with Au-NP ink dilution ratio 1:16 shows ca. 25 NPs/µm^2^. Sensor H exhibits a factor of ca. 14 less NPs per µm^2^, which corresponds roughly to the dilution of the employed NP inks.

## 3. Results

### 3.1. TEM Characterization of Nanoparticles

All NPs were characterized by means of Transmission Electron Microscopy (TEM). For this, specific TEM grids were drop-coated with the NP solutions. The NPs’ sizes were determined directly from the TEM-graphs to gain an efficient statistical value for the NP diameter; more than 100 NPs were analyzed per NP batch. This TEM analysis presents the most direct and accurate technology to determine the average NP diameters. All NP batches showed approximately spherical particle shapes with average diameters of 5.3 ± 0.5 nm, 4.4 ± 0.5 nm and 3.3 ± 0.4 nm for the Au, Pd and NiPt material systems, respectively (Figure 6). For concentration analysis the GNP batches were characterized through thermogravimetric analysis (TGA). The particle concentrations were calculated using the bulk material mass densities; the particle volumes were obtained from TEM analysis. Here, for the NiPt particle system a balanced stoichiometry was tentatively assumed.

### 3.2. Sensor Performance Measurement Procedure

The response of the sensors was investigated in an automated measurement setup, which enabled precise control of the gaseous environment by use of flow controllers. For the sensing measurements, humidified synthetic air at three relative humidity (rh) levels (25%, 50%, and 75%; at 20 °C) was employed as background gas, whereas pulses of CO and VOC test gases were introduced at ppm-level concentrations. The VOC test gas consisted of a specific hydrocarbon gas mixture (=HC_mix_) of acetylene, ethane, ethene, and propene (each of them 500 ppm, 2000 ppm in total). Both the synthetic air as well as the test gases, CO and HC_mix_, were ready-to-use mixtures diluted in nitrogen from the company Linde Gas.

All 16 gas sensors integrated on a single Si-platform chip were characterized by means of simultaneous resistance measurements in a 4-point configuration at a constant total gas flow of 1000 sccm in a test procedure as follows. First, the platform chip is heated up to 300 °C operation temperature for 15 min in synthetic air (50% rh). Then, a 50 ppm CO test gas pulse is inserted for 5 min; this is followed by 15 min synthetic air (50% rh). Next, a subsequent 50 ppm HC_mix_ test gas pulse is inserted for 5 min, again followed by 15 min synthetic air (50% rh). This procedure is repeated two more times for 25% rh and 75% rh, respectively. Figure 7 shows a typical resistance measurement of a bare and an Au-NP functionalized SnO_2_ (black and red curves, respectively) during exposure to 50 ppm of CO and HC_mix_ at 300 °C operation temperature for 50%, 25%, and 75% rh of the background gas. To demonstrate the usability of our platform chips for sensor screening, we have focused on a 50 ppm concentration for both test gases CO, and HC_mix_, which is highly relevant for real life scenarios: This is in the range of the maximum work place concentration (“maximale Arbeitsplatz-Konzentration”–MAK) of 30 ppm for carbon monoxide [40]. For the HC_mix_ components acetylene, ethane, ethene, and propene, there is no MAK value established yet; however, the presence of ethane in ambient air in a concentration range 25–100 ppm is supposed to be carcinogenic. As expected, the resistance decreases in the presence of both test gases; the decrease in the case of HC_mix_ is considerably higher than in the case of CO.

The relative resistance changes due to the interaction with the test gas, i.e., the sensor response *S*, was calculated according to Equation (1):(1)$$S=\frac{R_{air}-R_{gas}}{R_{air}}$$
where *R*_gas_ is the sensor resistance in the presence of the test gas and *R*_air_ is the sensor resistance in pure synthetic air. Following Equation (1), the responses of all sensors for the test gases CO and HC_mix_ have been calculated.

### 3.3. Response of Bare and Au-, NiPt-, and Pd-NP Functionalized SnO_2_ Sensors towards CO and HC_Mix_

The sensor measurements have been performed for bare SnO_2_ sensors, and SnO_2_ sensors functionalized with Au-, Pt-, and NiPt-NPs, respectively. All platform chips were functionalized in the scheme, as shown in Figure 4; two sensors per platform chip are functionalized with the same NP-concentration. Thus, for the sensor response, the average value of both sensors has been calculated.

Figure 8 shows the response of bare SnO_2_ sensors and sensors functionalized with different Au-NP-concentrations (1:1, 1:2, 1:4, 1:8, 1:16) towards 50 ppm CO and 50 ppm HC_mix_ at 300 °C operation temperature for 25%, 50%, and 75% rh. The CO and HC_mix_ sensor responses are significantly increased through the Au-NP functionalization and reach a maximum of ca. 10% and 38% for 25% rh, respectively, for the 1:1 ink concentration, as compared to 4% (CO) and 22% HC_mix_ for the bare SnO_2_. The response for the HC_mix_ is always significantly higher than for CO. With decreasing NP-concentration, the responses decrease again, even below the value of the bare SnO_2_ sensor.

Similar to Figure 8, Figure 9 shows the response of bare SnO_2_ sensors and sensors functionalized with different NiPt-NP-concentrations (1:1, 1:2, 1:4, 1:8, 1:16). The CO and HC_mix_ sensor responses are significantly increased through the NiPt-NP functionalization; the response towards CO reaches a maximum of ca. 18% for the 1:1 ink concentration for 25% rh, as compared to 2% for the bare SnO_2_. The response towards HC_mix_ reaches a maximum of ca. 45% for the 1:4 ink concentration for 25%rh, as compared to 16% for the bare SnO_2_. Again, the response for the HC_mix_ is always higher than for CO. With decreasing NP-concentration, the responses decrease below the value of the bare SnO_2_ sensor. Please note that, for an ink-concentration of 1:1, the sensor response is almost independent of relative humidity.

Figure 10 shows the response of bare SnO_2_ sensors and sensors functionalized with different Pd-NP-concentrations (1:1, 1:2, 1:4, 1:8, 1:16). In contrast to the results for Au- and NiPt-NPs, the CO and HC_mix_ sensor responses are decreased through the Pd-NP functionalization. Again, the response for the HC_mix_ is always higher than for CO.

## 4. Discussion

Sensor functionalization with metallic NPs has a strong impact on the sensor performance. All NPs except Pd increase the responses of the sensors towards CO and HC_Mix_, as compared to the bare SnO_2_ sensors. The results can be summarized as follows:
Au-NPs: the sensor responses for CO and HC_mix_ are increased, both showing a similar dependence on the NP concentration and reaching a maximum for a 1:1 Au-NP ink concentration.NiPt-NPs: the sensor responses for CO and HC_mix_ are significantly increased, show a clear dependence on the NP concentration, and reach a maximum around the 1:1 NiPt-NP ink concentration for the CO, and between the 1:2 and 1:4 NiPt-NP ink concentration for the HCmix. The dip of the CO response for a 1:2 NiPt-NP concentration cannot be explained but has been reproducible in all investigated samples.Pd-NPs: the results are significantly different, the sensor responses both for CO and HC_mix_ are decreased through the Pd-NP functionalization. Again, there is a dip for both responses for a 1:2 Pd-NP concentration, which cannot be explained, but has been reproducible in all investigated samples.


The responses of the bare SnO_2_ sensors differ significantly between different platform chips. This is mostly due to the spray pyrolysis deposition technology, where the target thickness is 50 nm. While the SnO_2_ films are uniform on a single 2 × 2 cm^2^ sized platform chip, the thickness considerably varies from sample to sample in a range of ±10 nm, which has a significant impact on the response. The thinner the SnO_2_ films, the higher the response; as observed in previous measurements, thickness fluctuation of ±10 nm results in a change of the response by a factor of 2–3.

In order to better reveal the influence of the NP functionalization, we have thus normalized the responses of Au, Pd, and NiPt-NPs functionalized sensors (for 50% rh) to the response of the bare SnO_2_ sensors, which are the reference sensors per platform chip (Figure 11). Towards CO, the Au- and NiPt-NPs cause the highest response for a 1:1 NP concentration. Pd-NPs also increase the response. There is this significant decrease for 1:2 concentrations for all types of NPs, in particular for the NiPt-NPs. Then, the response increases again (1:4), further increases for Pd, but decreases for NiPt and Au (1:8) and decreases to a minimum for lower concentrations (1:16). Towards the HC_mix_, the NiPt- and Au-NPs result in increased response, with maximum for NiPt-NPs at 1:2 concentrations, and for Au-NPs for 1:1 concentration. Pd-NPs partly increase the response towards CO but decrease the response towards HC_mix_ for all concentrations. For lower concentrations (1:16), the response of the functionalized sensors is significantly lower than that of the bare SnO_2_ sensors.

We cannot exclude organic contamination of the sensor surface due to the solvents and ligands used for the NP inks. However, in this case, all NP-functionalized sensors should be contaminated in the same way, because all of them are exposed to the solvents and ligands. Hence, we conclude, that the achieved performance improvement is clearly due to the influence of the NPs.

The physical/chemical mechanism of the NPs employed in our study cannot be clearly determined at the moment. In general, two possible mechanisms are proposed to account for the effects the catalytic NPs have on the surface reaction. The first mechanism, chemical sensitization, is characterized by the so-called spill-over effect. The adsorption of the gas molecule on the NP surface leads to its activation or dissociation. Then, the activated gas molecules migrate (=“spill over”) to the MOx surface and react with the adsorbed oxygen, leading to a change in the electrical conductivity of the MOx film. The second mechanism, electronic sensitization, is based on the direct exchange of electrons between the oxidized NPs and the MOx film. The reaction of the gas molecule with the NP surface changes the oxidation state of the NP and, as result, the electrical conductivity of the MOx film. For the NPs employed in our study, the most probable mechanism seems to be the spill-over effect, which was found to change the response of MOx based gas sensors for Au-NPs [41], Pd-NPs [42] and bimetallic NPs [43,44]. However, to fully understand the exact mechanisms, further investigations and specific surface characterization techniques (e.g., DRIFT, XPS, Kelvin-Probe microscopy) are necessary to explain the complex reaction mechanisms in detail.

## 5. Conclusions and Outlook

We have shown that the functionalization of ultrathin SnO_2_-film based conductometric sensors with metallic NPs is a powerful technology for improving the selectivity behaviors. Au-, NiPt-, and Pd-NPs with average diameters of 5.3 nm, 3.3 nm, and 4.4 nm, respectively, have been synthesized; octane/1-octanol mixtures were chosen as solvents for the NP inks to assure particle stability. For functionalization of the SnO_2_-based thin film sensors, the NP inks have been printed by ESJET technology, which provides well-controlled deposition of very small droplet volumes on the sensors. Different dilutions of the NP inks have been used to vary the NP-densities between some 25 NPs/µm^2^ and 350 NPs/µm^2^. The achieved results clearly demonstrate that the sensor responses can be tuned by using different types of NPs and different NP concentrations.

In this study, we have limited the NP functionalization to ultrathin SnO_2_ films as sensor structures. In the next step, we will employ our platform chip to increase the number of base materials (CuO and ZnO sensing films deposited by our own spray pyrolysis technology) and also a higher variation of nanoparticle materials (e.g., Pt, Ag, Cu, mono-, bi-, trimetallic and variations of NP characteristics, like size, ligands, etc.). In this study, we have focused on the Ni_0.3_Pt_0.7_ NPs stoichiometry due to the well-established synthesis procedure, but it is plausible and of high interest to investigate other NiPt ratios. This, however, requires the development of other synthesis routes in order to provide stable solutions for the ES-JET/ink-JET technologies. This work is in progress.

We are convinced that our screening platform technology can significantly contribute to optimizing chemical sensors due to its high flexibility, high stability (e.g., temperature, chemicals) and possibility of fast testing of various materials on one chip. While we have identified the range of optimum areal density of NP functionalization, the diameter of the NPs needs further investigation, because this will certainly influence the sensor response. Moreover, sequential functionalization of a single sensor with different types of NPs (e.g., mixture of Au and Pd NPs) is also plausible to further optimize the sensor performance.

The functionalization of MOx sensors with different types of metallic NPs provides a huge parameter space for optimizing chemical sensor devices. We are convinced that functionalizing the MOx-based materials with different metal NPs will enable a sensitive and selective multi-sensor system. Our platform chip will accelerate the research on different sensing nanomaterials to find the optimal combinations of MOx films and NP functionalization towards this goal.

Recently we have reported on CMOS-integrated 8 × micro-hotplate (µhp) arrays implementing 16 sensor devices on a single chip [26]. This chip will be employed to realize a multi-gas sensor device, where each of the sensors is optimized by specific NP functionalization to a specific target gas. As shown in Figure 12, different MOx films will be processed on the 8 µhps by means of subsequent spray pyrolysis deposition, photolithography, and MOx selective dry and wet-chemical etching. Each of the µhps implements two identical sensor films.

Next, the sensors will be individually functionalized with the ESJET printing technology: by using different types of NPs and various NP concentrations, the sensitivity of one sensor pair towards a specific target gas will be increased with one type of NP, while simultaneously decreased for another sensor pair with another type of NP. By proper combination of MOx type and type of NP (exemplarily shown for 4 µhps), we will be able to optimize the selectivity performance of all sensor pairs. This would result in a CMOS integrated gas sensor device capable of simultaneous detection of several gases. Both the spray pyrolysis technology for deposition of the MOx gas sensing films and the highly flexible ESJET NP functionalization could be performed on a 6” wafer scale. This paves the way toward mass production of such multi-gas sensor devices. This work is presently in progress.

## Figures and Tables

**Figure 1 sensors-24-05565-f001:**
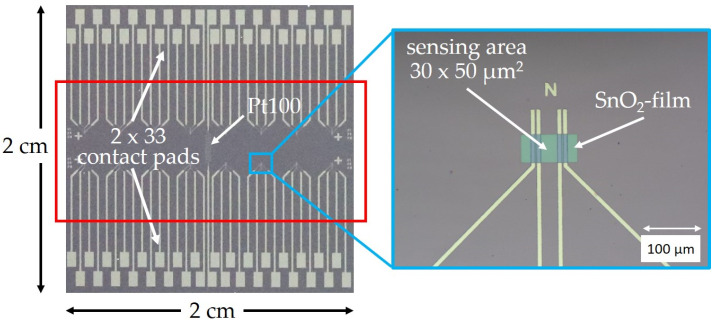
2 × 2 cm^2^ sized Si-based platform chip integrating 16 sensor devices. The chip enables conductive sensor measurement in a 4-point measurement configuration and exhibits 64 contact pads for the gas sensors; 2 additional pads are required for the Pt100-like temperature sensor. The red rectangle designates the “hot area” heated by the Au-coated Cu-block underneath. The insert shows a final processed 100 × 50 µm^2^ sized SnO_2_ film; the active sensor area between the electrodes measures 30 × 50 µm^2^.

**Figure 2 sensors-24-05565-f002:**
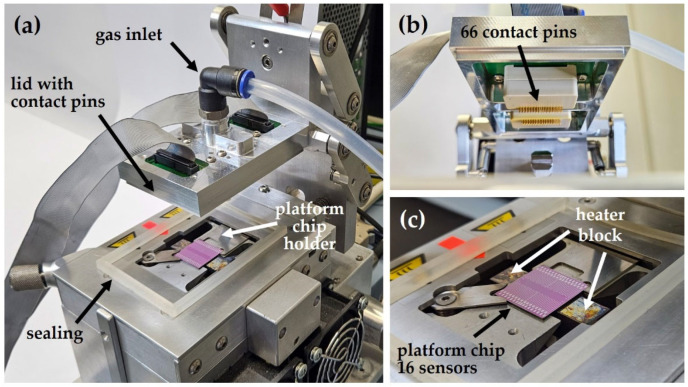
(**a**) Measurement setup (“open state”) consisting of a holder for the platform chip and a lid with gas inlet, electrical connectors, and contact pins. (**b**): Front view of the lid, which holds two prober heads with a total of 66 contact pins. (**c**) The platform chip is placed on a massive heater block (fabricated with copper, coated with gold) which heats the central part of the chip from underneath up to 350 °C. As soon as the lid is closed, the pins contact 66 Ti/Pt pads on the platform chip and enable simultaneous characterization of all 16 sensor structures in parallel.

**Figure 3 sensors-24-05565-f003:**
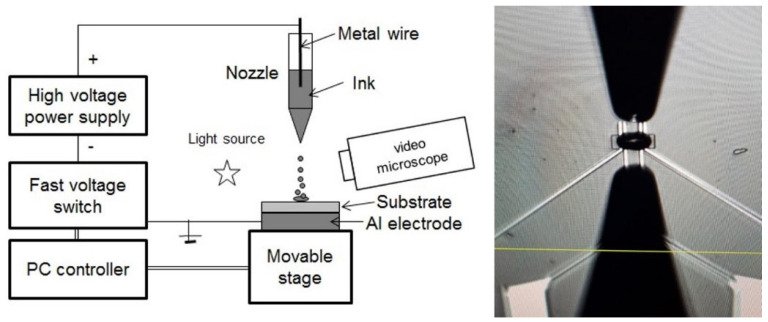
**Left**: Schematic of an ESJET system. A capillary (“emitter”) with the printing ink and a needle electrode is positioned a short distance above the grounded substrate. **Right**: ESJET nozzle positioned above a single SnO_2_ sensor film on the Si-platform chip; the NP-dot diameter matches the active area of the sensor films, which measures 30 × 50 µm^2^.

**Figure 4 sensors-24-05565-f004:**
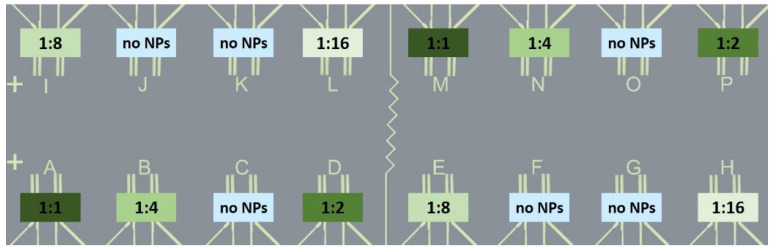
Printing scheme of the NP solutions with different NP concentrations (1:1, 1:2, 1:4, 1:8, 1:16) on the SnO_2_ sensor films on the Si-platform chips. Two sensors, respectively, are functionalized with the same NP concentration; six SnO_2_ sensor films are not functionalized and form the reference sensors.

**Figure 5 sensors-24-05565-f005:**
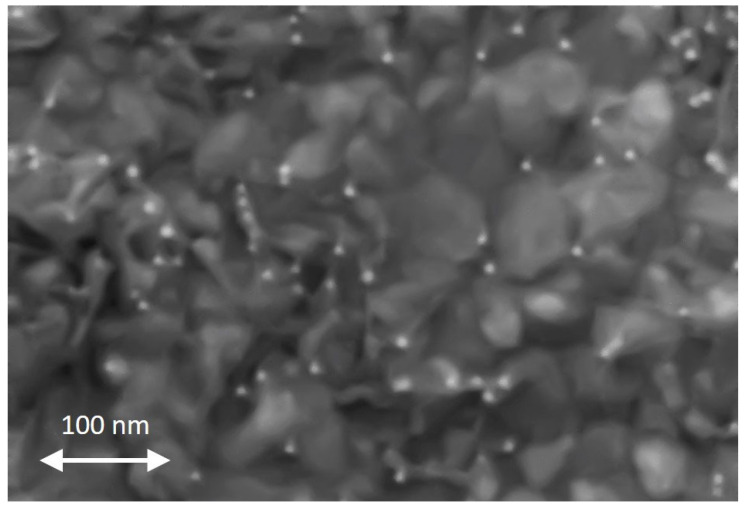
SEM graph of Sensor A, which is functionalized with Au-NPs; the Au-NPs can be clearly seen as white dots.

**Figure 6 sensors-24-05565-f006:**
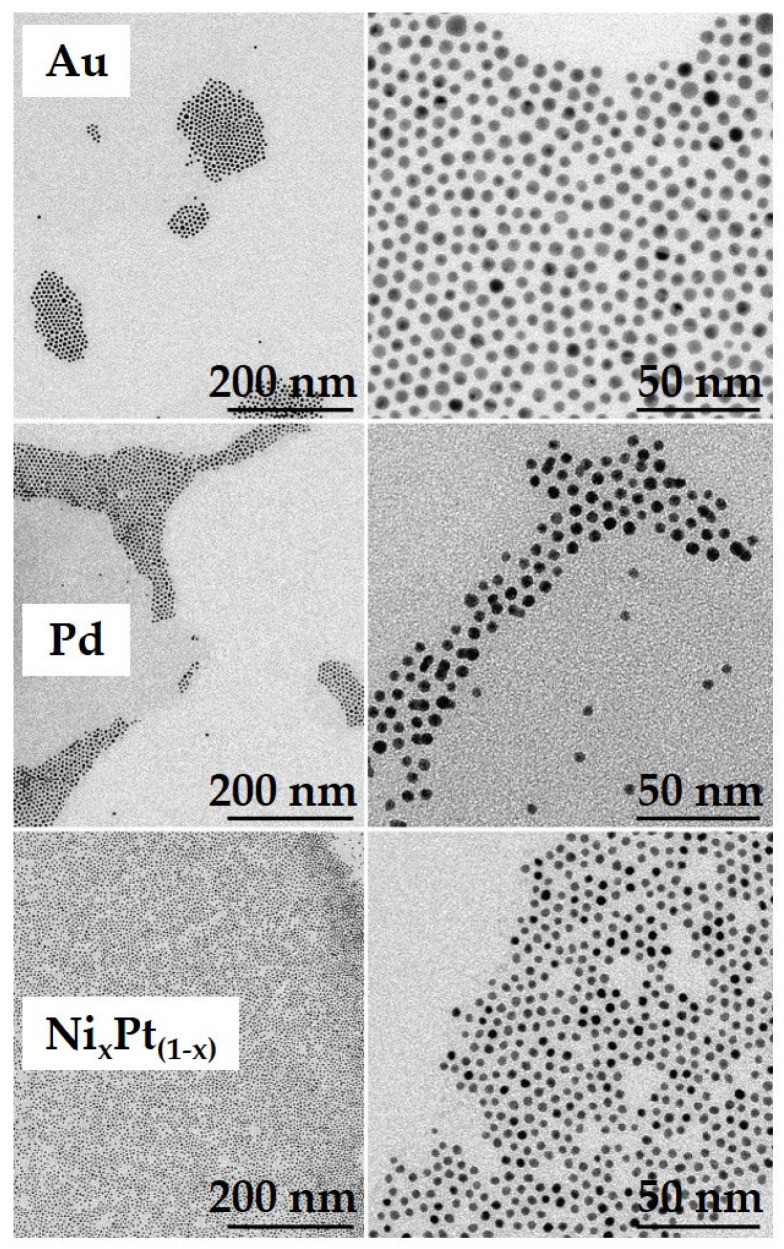
Transmission electron microscopy images of the noble metal NP batches used for sensor functionalization. Scale bars: 200 nm (**left** column), and 50 nm (**right** column). All particle batches showed approximately spherical particle shapes with average diameters of 5.3 ± 0.5 nm, 4.4 ± 0.5 nm, and 3.3 ± 0.4 nm for the Au, Pd, and NiPt material systems, respectively.

**Figure 7 sensors-24-05565-f007:**
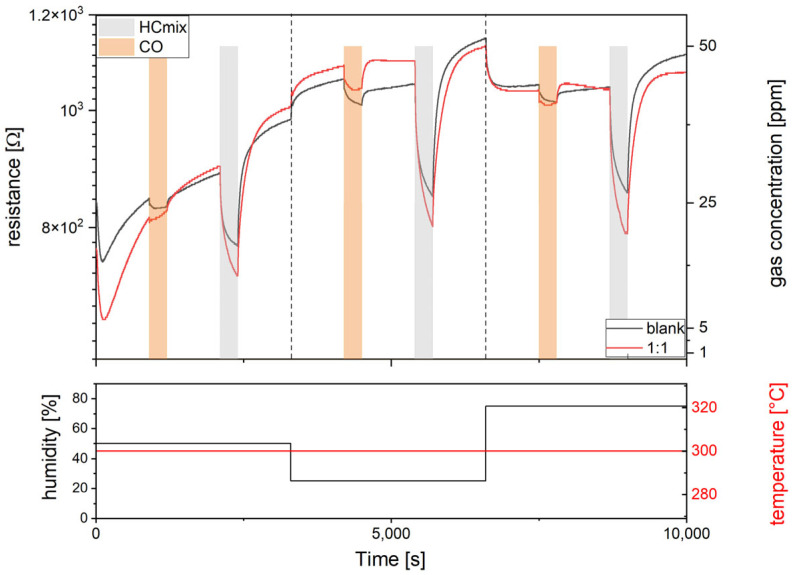
Typical resistance behavior of bare and an Au-NP functionalized SnO_2_ sensors (black and red curves, respectively) during exposure to 50 ppm of CO and HC_mix_ at 300 °C operation temperature (bottom red) at 50%, 25% and 75% humidity levels (bottom black).

**Figure 8 sensors-24-05565-f008:**
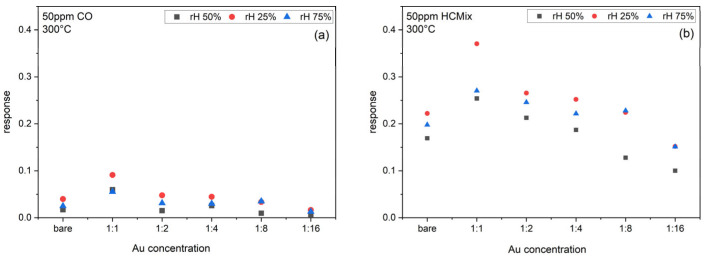
Response of bare SnO_2_ sensors and SnO_2_ sensors functionalized with different concentrations of Au-NP inks towards (**a**) 50 ppm CO, and (**b**) 50 ppm HC_mix_ at 300 °C operation temperature.

**Figure 9 sensors-24-05565-f009:**
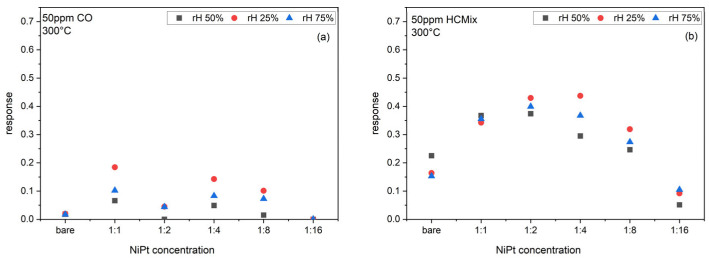
Response of bare SnO_2_ sensors and SnO_2_ sensors functionalized with different concentrations of NiPt-NP inks towards (**a**) 50 ppm CO, and (**b**) 50 ppm HC_mix_ at 300 °C operation temperature.

**Figure 10 sensors-24-05565-f010:**
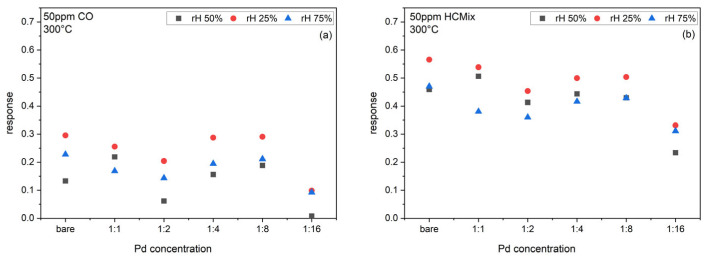
Response of bare SnO_2_ sensors and SnO_2_ sensors functionalized with different concentrations of Pd-NP inks towards (**a**) 50 ppm CO, and (**b**) 50 ppm HC_mix_ at 300 °C operation temperature.

**Figure 11 sensors-24-05565-f011:**
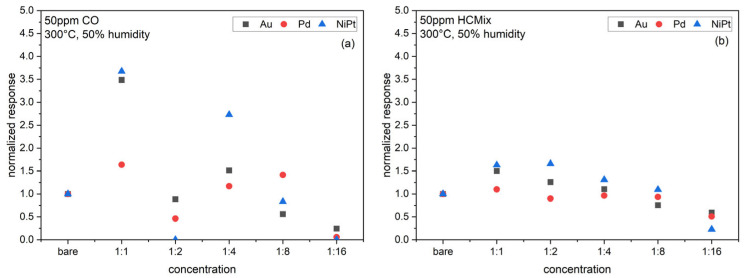
Response of sensors functionalized with Au, Pd, and NiPt-NPs with different concentrations towards (**a**) 50 ppm CO, and (**b**) 50 ppm HC_mix_ at 50% rh and 300 °C operation temperature normalized to the response of bare SnO_2_ sensors.

**Figure 12 sensors-24-05565-f012:**
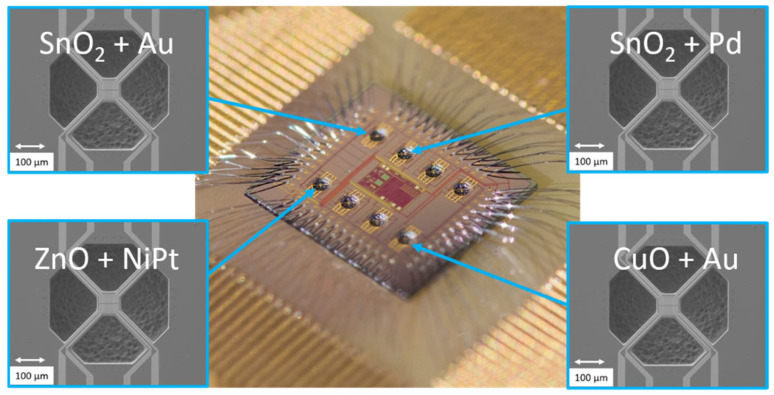
Proposed CMOS-integrated multi-sensor device. By using proper combinations of MOx film, and types of NPs on the 8 µhps (exemplarily shown for 4 µhps), we will realize a 5 × 5 mm^2^ sized multi-gas sensor device capable of simultaneous detection of several target gases.

**Table 1 sensors-24-05565-t001:** Dilution ratio, volume of the NP base solution, octane, and 1-octanol, respectively, for a total NP-ink volume of 5 mL. Resulting NP concentrations for the Au, NiPt, and Pd particle system.

Ink Lot.	V (NP Solution)	*V* (Octane)	*V*(1-Octanol)	*V*(Ink)	Au	NiPt	Pd
	[mL]				NP Ink Concentration [M]		
**1**	1	0	4	5	1.22 × 10^−6^	1.70 × 10^−6^	3.19 × 10^−6^
**1/2**	0.5	0.5	4	5	6.08 × 10^−7^	8.52 × 10^−7^	1.60 × 10^−6^
**1/4**	0.25	0.75	4	5	3.04 × 10^−7^	4.26 × 10^−7^	7.98 × 10^−7^
**1/8**	0.125	0.875	4	5	1.52 × 10^−7^	2.13 × 10^−7^	3.99 × 10^−7^
**1/16**	0.0625	0.9375	4	5	7.59 × 10^−8^	1.07 × 10^−7^	2.00 × 10^−7^

## Data Availability

The data presented in this study are available on request from the corresponding author.

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
