# Peer review of "Development of a Screening Platform for Optimizing Chemical Nanosensor Materials"

_sensors, 2024, doi:10.3390/s24175565_

Round 1

Reviewer 1 Report

Comments and Suggestions for Authors

The authors of the article demonstrate a very expensive method of modifying a fairly studied tin oxide to improve gas-sensitive properties. However, the gas-sensitive properties have not been studied enough to make a definite statement. that modification with precious metal particles will allow to create a promising gas sensor

1) the authors have not studied the effect of gas concentration

2) There is no justification for the operating temperature of the sensor

3) there is no data on the study of selectivity

4) the expediency of modification by noble metal particles is insufficiently substantiated

5) the response value of the sensor modified with metals is insufficient to draw conclusions about the prospects of such gas sensors

Author Response

The authors of the article demonstrate a very expensive method of modifying a fairly studied tin oxide to improve gas-sensitive properties. However, the gas-sensitive properties have not been studied enough to make a definite statement. that modification with precious metal particles will allow to create a promising gas sensor:

Comments 1: the authors have not studied the effect of gas concentration

Response 1: The aim of this paper was to develop a platform chip, which improves and accelerates the investigation of nanomaterials for gas sensing. Therefore, we tested in this work only for one gas concentration (50ppm HCmix and CO close to the MAK values, see subsection “3.2 Sensor performance measurement procedure”), where we expect to get sensor response. However, in further work we will perform a full characterization of the investigated materials including different gas concentrations, sensor operation temperatures and selectivity. We have changed the title and improved the abstract and introduction to make this goal clearer.

Comments 2: There is no justification for the operating temperature of the sensor

Response 2: Thank you for pointing this out. We added this information in the Introduction: In literature it was shown that those parameters are representative and promising sensing parameters for SnO2-based gas sensors. [36–39].

Comments 3: there is no data on the study of selectivity

Response 3: This was not the aim of this work. A study on the selectivity of different metallic nanoparticle functionalized MOx will be investigated in future works.

Comments 4:  the expediency of modification by noble metal particles is insufficiently substantiated

Response 4: The effect of metallic nanoparticle functionalization has been sufficiently reported in the literature over the last years. The performance of SnO2-based devices can be enhanced by the addition of noble metals as catalysts to the MOx [19,20]. Tailoring the response of MOx sensors by surface functionalization with metallic nanoparticles such as Au, Pd, or Pt, is a most promising approach to achieve a high degree of selectivity [21–24].

Comments 5:  the response value of the sensor modified with metals is insufficient to draw conclusions about the prospects of such gas sensors

Response 5: The purpose of this study was to show the screening capability of our platform system and show the first results of the screening of the metallic NPs. However, as demonstrated in Fig. 9 we have improved the response of a bare SnO2 by factor of 2 towards HCmix through functionalizing the sensors with a 1:2 NiPt solution. As discussed in Conclusion and Outlook, the response of the sensor can be clearly improved by metallic NPs functionalization, but this will be further optimized within our future studies.

Reviewer 2 Report

Comments and Suggestions for Authors

The work is very well presented and very interesting.

The methodology is clear and the results are discussed.

Only some minor points to check

-In abstract line 21 and in introduction line 87: please modify "ethan ethen propen" by ethane, ethene and propene

-In introduction, line 76: "200 mm wafer size" looks strange, maybe 20x20 mm2 ?

-In materials and methods, line 109: 66 pads = 2x32+2 instead of 33

-In table 1 (caption and table) please remplace "octan" by octane

-I also think that NP solution is in octane but it is not clearly mentionned. If it is not the case, the proportion 20/80 reported lines 175-176 is wrong, so this point is to check. The volumes indicated in the table should clearly indicate this point also.

-In results part, according to lines 279-280, the colors in figure 7 for CO and HCMix injections are reversed.

-It is also important to give the composition of the hydrocarbon gas mixture (proportion of each hydrocarbons).

Author Response

Comments and Suggestions for Authors

The work is very well presented and very interesting.

The methodology is clear and the results are discussed.

Only some minor points to check:

Comments 1: In abstract line 21 and in introduction line 87: please modify "ethan ethen propen" by ethane, ethene and propene

Response 1: We changed it.

Comments 2: In introduction, line 76: "200 mm wafer size" looks strange, maybe 20x20 mm2 ?

Response 2: Thank you for the comment. We made it clearer in the introduction.

Comments 3: In materials and methods, line 109: 66 pads = 2x32+2 instead of 33

Response 3: We changed the wording to make it clearer.

Comments 4: In table 1 (caption and table) please remplace "octan" by octane

Response 4: We changed it.

Comments 5: I also think that NP solution is in octane but it is not clearly mentionned. If it is not the case, the proportion 20/80 reported lines 175-176 is wrong, so this point is to check. The volumes indicated in the table should clearly indicate this point also.

Response 5: We added that the solutions are in octane.

Comments 6: In results part, according to lines 279-280, the colors in figure 7 for CO and HCMix injections are reversed.

Response 6: Thank you. We have changed it.

Comments 7: It is also important to give the composition of the hydrocarbon gas mixture (proportion of each hydrocarbons).

Response 7: We missed to mention it and have added it now in the subsection 3.2 Sensor performance measurement procedure.

Reviewer 3 Report

Comments and Suggestions for Authors

The authors have analyzed the performance of doped SnO2 sensors with different dopant (Au, Pd and NiPt) concentrations using a sensor screening platform. Some corrections need to be done to enable better understanding of the focus of the work.

The title is to a degree too general and needs to be more focused on what was actually done in this work. The screening platform was designed in previous work by some of the authors. The abstract is also too general and needs to contain some results of the optimization process. It is not enough to say that parameters can be tailored. The authors have analyzed 3 different dopants, different solution concentrations and also the influence of relative humidity. This all needs to be mentioned in the abstract and also the Introduction. The authors need to state more clearly what was the focus of the work and describe this more clearly and precisely.

The experimental section contains some results of nanoparticle characterization, maybe they could be moved to the results section. The authors give a figure in the conclusion, maybe this could be moved to the discussion.  

How did the authors determine x for the NiPt mixture? They mention 0.3 in the Introduction, yet in the experimental section it is only x?

Author Response

The authors have analyzed the performance of doped SnO2 sensors with different dopant (Au, Pd and NiPt) concentrations using a sensor screening platform. Some corrections need to be done to enable better understanding of the focus of the work.

Comments 1: The title is to a degree too general and needs to be more focused on what was actually done in this work. The screening platform was designed in previous work by some of the authors. The abstract is also too general and needs to contain some results of the optimization process. It is not enough to say that parameters can be tailored. The authors have analyzed 3 different dopants, different solution concentrations and also the influence of relative humidity. This all needs to be mentioned in the abstract and also the Introduction. The authors need to state more clearly what was the focus of the work and describe this more clearly and precisely.

Response 1: We changed the title and the abstract to state clearer what we have done.

Comments 2: The experimental section contains some results of nanoparticle characterization, maybe they could be moved to the results section. The authors give a figure in the conclusion, maybe this could be moved to the discussion.  

Response 2: Thank you for the comment. We added the subsection 3.1 TEM characterization of nanoparticles in the results and moved the NP characterization there.

Comments 3: How did the authors determine x for the NiPt mixture? They mention 0.3 in the Introduction, yet in the experimental section it is only x?

Response 3: We added the following explanation in subsection 2.2. Synthesis of nanoparticles: The composition of the NixPt1-x nanoparticles was determined via energy-dispersive X-ray analysis (EDX) using a scanning electron microscope (SEM). Therefore, the purified nanoparticles dispersed in hexane were dropcasted on a silicon wafer. The resulting nanoparticle film was examined in a Zeiss LEO Gemini 1550. The silicon drift detector used was an Oxford Instruments Ultim Max 100. The composition determined was within the range described in the literature.

Round 2

Reviewer 1 Report

Comments and Suggestions for Authors

The authors have changed the title and scope of the work. However, the introduction does not meet the stated goals. Currently, the introduction is devoted to describing what the authors have done in the work. However, in the introduction, an overview of existing solutions in the field of creating multisensory platforms is expected. We need to understand why such platforms are needed. Most often, it is used to detect gases in mixtures. It is also necessary to describe the original platform developed at the Karlsruhe Institute of Technology  (Germany) also functioning on the basis of tin oxide.

At present time the purpose of the work is not clear. Why we need in such platforms?

From conclusion: "We have shown that the functionalization of ultrathin SnO2-film based conductometric sensors with metallic NPs is a powerful technology for improving the selectivity behaviors". What is the role of the platform? I see only the role of the material. 

Author Response

Comment 1: The authors have changed the title and scope of the work. However, the introduction does not meet the stated goals. Currently, the introduction is devoted to describing what the authors have done in the work. However, in the introduction, an overview of existing solutions in the field of creating multisensory platforms is expected. We need to understand why such platforms are needed. Most often, it is used to detect gases in mixtures. It is also necessary to describe the original platform developed at the Karlsruhe Institute of Technology  (Germany) also functioning on the basis of tin oxide.

At present time the purpose of the work is not clear. Why we need in such platforms?

From conclusion: "We have shown that the functionalization of ultrathin SnO2-film based conductometric sensors with metallic NPs is a powerful technology for improving the selectivity behaviors". What is the role of the platform? I see only the role of the material. 

Response 1: Thank you very much for your comments!

The platform chip you referred to is a temperature-gradient-based sensor chip (KAMINA, Karlsruhe Institute of Technology), which can be already used as multisensory platform device. In our case we developed the platform chip not as a multi-sensor device but as a tool for fast, cheap and accelerated material evaluation to optimize the gas sensing performance. As an example, which is presented in the paper, we employed the ESJET technology for functionalization of the SnO2 gas sensing material with metallic nanoparticles. Presently, in addition to the base material SnO2, we are using this method to evaluate a variety of other gas sensing materials.

Hence, the major role of our platform chip is to screen different materials and combinations thereof (e.g. SnO2 with different NPs, we are also using different MOx base materials, not included in this work), to find the best materials for specific target gases. In the next step we will use the identified materials for the multi-sensor array chip, as presented in section 5. "Conclusions and Outlook" in Figure 12.

.

Reviewer 3 Report

Comments and Suggestions for Authors

The authors have addressed all the comments made in the review, answered them and significantly improved their work. I recommend publication in its present form.

Author Response

Comment 1: The authors have addressed all the comments made in the review, answered them and significantly improved their work. I recommend publication in its present form.

Response1: Thank you very much!